# The spillover effects of Medicare's comprehensive care for joint replacement (CJR) model in California

**Narae Kim** [1]*, **Mireille Jacobson** [1,2]

**1** Leonard Davis School of Gerontology, University of Southern California, Los Angeles, California, United States of America, **2** Leonard D. Schaeffer Center for Health Policy & Economics, University of Southern California, Los Angeles, California, United States of America

* naraekim@usc.edu

## Abstract

The Comprehensive Care for Joint Replacement (CJR) model, a bundled Medicare payment for lower extremity joint replacement (LEJR), was initially randomized across the United States, providing a unique opportunity to study the broad impact of this alternative payment model. This study aimed to determine the spillover effects of the CJR model on older patients in California covered outside of the traditional Medicare program. The study analyzed hospitalizations for hip and knee joint replacement in California between January 2014 and December 2017 from the California Patient Discharge Dataset. The study used event study and difference-in-differences models to estimate changes in discharge-related outcomes in hospitals in treated and control areas before versus after CJR implementation (April 2016). Main outcomes were hospital length of stay and home discharge rates. All LEJR patients admitted to the treated or control hospitals were included in the study regardless of their primary payers. Of 312,914 analyzed LEJR hospitalizations (mean [SD] age, 68.3 [11.3] years; 189,575 [60.6%] women; 15,374 [4.9%] black), 113,590 (36.3%) were covered by traditional Medicare (TM), 83,277 (26.6%) were covered by Medicare Advantage (MA), and 116,047 (37.1%) were without Medicare coverage. After program implementation, TM and non-Medicare patients in treated hospitals experienced reductions in length of stay (-4.0% & -1.0%, p<0.05) and TM, MA and non-Medicare patients in treated hospitals experienced increases in home discharge rates (3.4%, 4.7% & 2.3%, p<0.001) relative to patients in untreated hospitals. CJR affected health care for non-targeted populations. Evaluating the program based on traditional Medicare beneficiaries alone does not capture the entire effect of the program on older adults.

## Introduction

Lower Extremity Joint Replacement (LEJR) is one of the most common treatments received by Medicare beneficiaries, accounting annually for about 5% of both admissions and inpatient spending [1]. In April 2016, the Center for Medicare and Medicaid Services (CMS) introduced the Comprehensive Care for Joint Replacement (CJR) model to evaluate whether a bundled payment could reduce the overall cost of care for LEJR in traditional Medicare. In place of

**Data availability statement:** This study analyzed California's Patient Discharge Data

(PDD) provided by the California Department of Health Care Access and Information (HCAI) for all hospitalizations for major hip or knee joint replacement (MS-DRG 469 and 470) from 2014 to 2017. The dataset contains comprehensive information on patients admitted to hospitals in California, such as the admitting hospital, admission and discharge dates, patient's age, race, and primary diagnosis and discharge disposition (e.g., discharged to home or a skilled nursing facility), irrespective of the payer. To use the dataset, the authors obtained approval from the Committee for the Protection of Human Subjects (CPHS) at the California Health and Human Services. Protocol ID: 2022-205 Protocol Title: The Spillover Effects of Comprehensive Care for Joint Replacement (CJR) Model: A Study from California The authors confirmed that they received approval from CPHS to access the data. To gain access to the data, please follow the steps below: 1) Create a data request from HCAI data request portal (https://datarequest.hcai.ca.gov/csm) 2) Submit a protocol for IRB review from CPHS (https://www.cdii.ca.gov/committees-and-advisory-groups/committee-for-the-protection-of-human-subjects-cphs/) *Contact information for the California Department of Health Care Access and Information: Phone (916) 326-3800 Office 2020 West El Camino Avenue, Suite 800, Sacramento, CA 95833

**Funding:** This study was supported by a Haynes Lindley doctoral dissertation fellowship from the Haynes Foundation and grant 1R01HS026488-01A1 from the Agency for Healthcare Research and Quality. There was no additional external funding received for this study. The funders had no role in study design, data collection and analysis, decision to publish, or preparation of the manuscript.

**Competing interests:** The authors have declared that no competing interests exist.

fee-for-service payments, the CJR model provides a bundled payment for an episode of care and compensates hospitals that meet target spending and a minimum quality score via a reconciliation payment [2]. Importantly, the bundle includes not only inpatient care but also LEJR-related care received 90 days post-discharge [2].

Another distinctive feature of CJR is that it was implemented based on a randomization process. CMS randomly selected 67 Metropolitan Statistical Area (MSA) from across the country for mandatory participation in the payment model in 2016. All hospitals in these MSAs, with the exception of those participating in the Bundled Payments for Care Improvement (BPCI) initiative for LEJR treatment, were required to participate in CJR and were reimbursed for total hip and knee replacements based on a bundled payment. The mandatory nature of these bundles was controversial, and in 2018, hospitals in 33 MSAs were given the option to opt out of the program; the 34 MSAs that historically had the highest episode spending had to remain in the program [3].

To date, multiple studies have examined CJR and demonstrated its success in traditional Medicare. CJR reduced the average cost of an episode of care to Medicare [4–9]. Savings were driven primarily by reduced utilization of institutional long-term care; a larger proportion of patients were discharged to home, with or without services from home health agencies. Hospital readmission rates, ambulatory care utilization, mortality, and markers of the quality of care were unaffected [10,11].

As one of the largest payers, Medicare has the potential to broadly affect the market for health care. Some work has considered the "spillover" effects, or indirect effects of the policy beyond its intended target, of the CJR model [12,13]. Specifically, past work has analyzed the CJR model's effect on untargeted insured groups – Medicare Advantage (MA) patients, who are covered by private insurance plans that contract with the Medicare program, and patients with commercial insurance plans outside of Medicare. However, studies of the spillover effects of CJR have come to conflicting conclusions. Some studies have found that the share of MA patients discharged to institutional care after a hip or knee replacement decreased in treated relative to control MSAs [1,14]. A study of hospital discharge data from Florida, one of the highest spending areas in the country, found substantial spillover effects of CJR on home discharge rates for MA and commercially insured LEJR patients as well as for patients who went through non-LEJR procedures [15]. In contrast, two studies using claims data from the Health Care Cost Institute found no evidence of CJR spillovers to MA patients or commercially insured patients receiving LEJR [16,17].

In this study, we examined both the direct and spillover effects of CJR using hospital discharge data from California. We examined changes in health care service utilization associated with the program on three patient groups: traditional Medicare (TM) patients, who were subject to CJR, as well as Medicare Advantage (MA) and non-Medicare patients, both of whom were not subject to CJR. Since California accounts for over 10% of Medicare beneficiaries in the United States [18] with considerable racial, ethnic, and socioeconomic diversity, understanding the impact of CJR in the state is of direct importance. Given the conflicting literature, the primary goal of this work was to contribute to our understanding of the spillover effects of CJR on untargeted populations.

## Materials and methods

### Data

This study analyzed California's Patient Discharge Data (PDD) provided by the California Department of Health Care Access and Information (HCAI) for all hospitalizations for major hip or knee joint replacement (MS-DRG 469 and 470) from 2014 to 2017. The dataset

contains comprehensive information on patients admitted to hospitals in California, such as the admitting hospital, admission and discharge dates, patient's age, race, and primary diagnosis and discharge disposition (e.g., discharged to home or a skilled nursing facility), irrespective of the payer. The authors had no access to information that could identify individual participants during or after data collection.

The study protocol was approved by the State of California's Committee for the Protection of Human Subjects (CPHS) and the University of Southern California's Institutional Review Board. California's CPHS waived informed consent based on the infeasibility of obtaining consent as well as the minimal risk of causing harm. The results were reported using the Strengthening the Reporting of Observational Studies in Epidemiology (STROBE) reporting guideline.

## Study population

Hospitalizations for three different patient groups were studied: traditional Medicare (TM), Medicare Advantage (MA), and non-Medicare patients. We limited Medicare beneficiaries to those ages 65 and over, assuming that those who are under the age of 65 but receive Medicare benefits would have other medical conditions that could bias our estimation. For the non-Medicare patient group, we limited the sample to those ages under 65 to only include non-targeted younger adults for a clearer comparison. In California, three MSAs – San Francisco-Oakland-Hayward, Modesto, and Los Angeles-Long Beach-Anaheim – among 26 MSAs were mandated to participate in the CJR model in 2016 and 2017. Thus, the treated group included LEJRs at hospitals in the three participating MSAs regardless of their primary payer. The control group included those at hospitals in the other 23 MSAs. LEJR hospitalizations from 37 hospitals participating in the BPCI initiatives were excluded from both treated and control groups. In addition, hospitalizations without information on primary payer were excluded.

## Outcomes

Our primary outcomes of interest were inpatient length of stay and discharge home, two proxies for LEJR-related healthcare service utilization. Multiple prior studies have investigated these two outcomes as the primary expected sources of Medicare savings from LEJR [4,5,7–9]. Therefore, changes in these outcomes were analyzed to measure direct and indirect effects of the CJR model. For inpatient length of stay, we used the HCAI's recommended adjusted length of stay variable, which replaced 0 days in the length of stay variable with 1 to give value to patients who were admitted and expected to stay overnight but discharged home on the same day [19]. We further applied a logarithmic transformation to the adjusted length of stay to handle the outcome's skewed distribution. To analyze discharge status, we created a binary indicator for home discharge (1 = being discharged to home; 0 = otherwise). Home discharge included self-care at home as well as care at home from an organized home health service organization or a hospice [20].

## Covariates

Covariates included both hospital and quarter-year fixed effects to control for time-invariant characteristics of each hospital and general trends in outcomes across California. We also controlled for demographic characteristics of patients that could possibly affect the health outcomes of interest; the demographic characteristics included age and its square, a binary indicator for female (1 = female, 0 = otherwise), a categorical variable for race/ethnicity and MS-DRG code indicators – an indicator for comorbidities.

## Statistical analysis

We estimated the impact of the CJR program using event study and difference-in-differences (DID) analysis. Both event study and DID analysis effectively control for time trends in pre-post assessments, a common health care policy evaluation method, by comparing treated and control groups before and after a treatment (i.e., policy implementation) [21]. While an event study shows temporal, relative changes in outcomes, a DID analysis provides an average, summary estimate of the changes the treated group experienced relative to the control group after policy implementation.

With the event study, we estimated quarter-year patient-level changes in the outcomes of interest – adjusted length of stay or home discharge rate – before and after CJR implementation. In (1), we show the equation for the event study:

$$Y_{ihmq} = \alpha_{hm} + \gamma_q +_q * \ Treated_m + X'_{ihmq}\beta + \varepsilon_{ihmq} \qquad (1)$$

where $Y_{ihmq}$ is the outcome of interest (log adjusted length of stay or discharge home) for patient $i$ treated in hospital $h$ in MSA $m$ and quarter-year $q$. The regression includes indicators for hospitals, $\alpha_{hm}$, to control for fixed differences in outcomes across hospitals and indicators for quarter-year $\gamma_q$, to flexibly control for general time trends as well as a set of patient characteristics included in the discharge dataset $X'_{ihmq}$. Our interest is in $_q$ the coefficients on the quarter-year fixed-effects interacted with the treatment indicator for whether the MSA where the hospital was located was randomized to the CJR treatment. We omitted the interaction term for the first quarter of 2016 such that estimates are normalized to the quarter before CJR took effect. Consequently, the coefficients show the temporal difference in outcomes between treated and control hospitals relative to the reference period and allow us to assess whether our difference-in-differences estimates capture a change in outcomes that is credibly related to CJR.

In the DID models, we estimated the differential change in outcomes after relative to before policy implementation in treated relative to control hospitals. In (B), we show the equation for the difference-in-differences analysis:

$$Y_{ihmq} = \alpha_{hm} + \gamma_q + \delta post_{mq} + X'_{ihmq}\beta + \varepsilon_{ihmq} \qquad (2)$$

where $Y_{ihmq}$, $\alpha_{hm}$, $\gamma_q$ and $X'_{ihmq}$ are as defined above and $post_{mq}$ is an indicator that captures the interaction between the post CJR period (Q2 of 2016 and later) interacted with an indicator for the treated hospitals (i.e., hospitals located in MSAs randomized to CJR). Note that the main post-period effect is subsumed in the quarter indicators and the main MSA effects are subsumed in the hospital indicators. Our key parameter of interest is the coefficient $\delta$ on $post_{mq}$, which captures the differential change in outcomes in hospitals randomized to CJR after the program was implemented compared to the change for hospitals not randomized to participate in CJR.

We conducted several sensitivity analyses to test the validity of our findings. As pre-policy trends for MA patients were generally not parallel, a condition for valid DID interpretation, we conducted analyses that added pre- and post-policy time trends to the event study and DID models for this group. Second, we employed a wild cluster bootstrap to address potential concerns about inference with a treatment (program participation) that was clustered at the MSA level but had only a small number of treated and control clusters. The wild cluster bootstrap offers a procedure based on the wild bootstrap method [22,23] but that takes into account clustering in the dataset with only a small number of clusters [24]. If a discrepancy exists, it suggests that the estimated standard errors from the main

analyses may have been understated, a common issue when using clustered standard errors in settings with few treated clusters [24]. Other sensitivity analyses included DID analyses (1) without any age restrictions on the sample, (2) that controlled for admission from an emergency department (ED) to control for unobserved health differences, (3) with an indicator for Medicaid eligibility for non-Medicare patients, a measure of socioeconomic status that could impact health outcomes and (4) that limited to the period between 2015 and 2017 and used propensity score weighting to ensure balance between patients in treatment and control MSAs. Statistical results were considered significant at a confidence level of 95%. We used Stata/MP 16.1 for all analyses. Data were collected and analyzed between December 2022 and July 2023.

## Results

The total number of hospitalizations for LEJR from January 2014 to December 2017 in California was 411,004. Across all LEJR hospitalizations, 146,662 (35.7%) had traditional Medicare as the primary payer (mean [SD] age, 73.7 [8.7] years; 93,249 [63.6%] women; 5,131 [3.5%] black) while 96,411 (23.5%) had Medicare Advantage (MA) (mean [SD] age, 74.2 [8.1] years; 62,210 [64.5%] women; 5,239 [5.4%] black), and 167,905 (40.8%) had non-Medicare sources as the primary payer (mean [SD] age, 59.7 [8.9] years; 91,824 [54.7%] women; 10,782 [6.4%] black) (S1 Table in S1 File).

After applying our sample restrictions – only including those ages 65 and over for TM and MA and those under 65 for Non-Medicare and excluding those admitted to hospitals participating in BPCI – the total number of hospitalizations included in the study was 312,914 (mean [SD] age, 68.3 [11.3] years; 189,575 [60.6%] women; 15,374 [4.9%] black). Among them, 113,590 (36.3%) were covered by TM, 83,277 (26.6%) were covered by MA, and 116,047 (37.1%) were without Medicare coverage. Among TM hospitalizations, 59,214 (52.1%) occurred after the program; 51,708 (45.5%) were from treated hospitals. Among MA hospitalizations, 37,757 (45.3%) were after the program; 41,943 (50.4%) were from treated. Among non-Medicare covered hospitalizations, 50,818 (43.8%) were after the program; 57,061 (49.2%) were from treated. Overall demographic characteristics of the patients in treated and control MSAs were not substantially different except by race. The proportion of non-Hispanic white patients was higher in the control MSAs (Table 1).

Plots of the unadjusted outcomes from January 2014 to December 2017 showed that gaps in outcomes between hospitals in treated and control MSAs generally narrowed after CJR implementation (S1 and S2 Figs in S1 File). Log adjusted length of stay was higher for TM and non-Medicare patients in hospitals in treated relative to control MSAs, but the gap narrowed after CJR implementation (S1 Fig in S1 File). Home discharge rates across all patient groups were higher in control MSAs before CJR implementation, but the gap narrowed and even reversed after CJR implementation (S2 Fig in S1 File).

Event study analyses showed quarter-year changes in the logged adjusted length of stay and home discharge rates in treated MSAs relative to control MSAs (Figs 1 and 2). Hospitals in treated MSAs had a relative decrease in logged adjusted length of stay after CJR model implementation among TM and non-Medicare patients (Fig 1). Only MA hospitalizations had a relative increase in the length of stay in the treated MSAs. However, the increase started before the CJR model implementation, suggesting the increase was not caused by CJR policy implementation. To assess this possibility, we conducted a supplemental analysis for MA patients that included separate linear pre- and post-policy trends; adjusting for the linear time trends, the MA-covered hospitalizations in treated MSA showed a relative decrease after policy implementation (S3 Fig in S1 File).

**Table 1. Characteristics of Traditional Medicare, Medicare Advantage and non-Medicare Patients Received Lower Extremity Joint Replacement in Treatment and Control MSAs in California from 2014 to 2017 (N = 312,914).**

| Characteristics | No. (%) | | | | | | | | | | | |
|---|---|---|---|---|---|---|---|---|---|---|---|---|
| | Traditional Medicare | | | | Medicare Advantage | | | | Non-Medicare | | | |
| | Treated MSA | | Control MSA | | Treated MSA | | Control MSA | | Treated MSA | | Control MSA | |
| | Pre-CJR | Post-CJR | Pre-CJR | Post-CJR | Pre-CJR | Post-CJR | Pre-CJR | Post-CJR | Pre-CJR | Post-CJR | Pre-CJR | Post-CJR |
| No. | 24,398 | 27,310 | 29,978 | 31,904 | 22,769 | 19,174 | 22,751 | 18,583 | 31,785 | 25,276 | 33,444 | 25,542 |
| Age, mean (SD) | 75.1 (7.3) | 75.0 (7.2) | 75.1 (7.4) | 75.0 (7.3) | 75.1 (7.2) | 75.1 (7.1) | 75.1 (7.2) | 75.1 (7.2) | 56.5 (6.8) | 56.8 (6.8) | 56.9 (6.3) | 57.0 (6.4) |
| Women | 15,807 (64.8) | 17,663 (64.7) | 19,004 (63.4) | 20,086 (63.0) | 14,814 (65.1) | 12,665 (66.1) | 14,517 (63.8) | 11,959 (63.4) | 17,379 (54.7) | 13,776 (54.5) | 18,080 (54.1) | 13,825 (54.1) |
| Race/ethnicity | | | | | | | | | | | | |
| White | 17,697 (72.5) | 19,841 (72.7) | 24,537 (81.9) | 26,022 (81.6) | 14,608 (64.2) | 11,966 (62.4) | 17,951 (78.9) | 14,553 (78.3) | 19,418 (61.1) | 15,209 (60.2) | 24,319 (72.7) | 18,185 (71.2) |
| Black | 982 (4.0) | 1,086 (4.0) | 544 (1.8) | 574 (1.8) | 1,577 (6.9) | 1,292 (6.7) | 765 (3.4) | 613 (3.3) | 2,961 (9.3) | 2,171 (8.6) | 1,596 (4.8) | 1,213 (4.8) |
| Hispanic | 2,529 (10.4) | 2,603 (9.5) | 2,880 (9.6) | 3,023 (9.5) | 4,448 (19.5) | 4,015 (21.0) | 2,928 (12.9) | 2,426 (13.1) | 6,239 (19.6) | 5,205 (20.6) | 5,359 (16.0) | 4,284 (16.8) |
| Asian/Pacific Islander | 1,688 (6.9) | 2,006 (7.3) | 970 (3.2) | 1,102 (3.4) | 1,667 (7.3) | 1,463 (7.6) | 722 (3.2) | 653 (3.5) | 1,750 (5.5) | 1,517 (6.0) | 1,050 (3.1) | 851 (3.3) |
| Native American/ Eskimo/Aleut | 31 (0.1) | 25 (0.1) | 84 (0.3) | 81 (0.3) | 43 (0.2) | 53 (0.3) | 44 (0.2) | 41 (0.2) | 77 (0.2) | 63 (0.3) | 161 (0.5) | 106 (0.4) |
| Other or unknown | 1,471 (6.1) | 1,749 (6.4) | 963 (3.2) | 1,102 (3.4) | 426 (1.9) | 385 (2.0) | 265 (1.2) | 218 (1.2) | 1,340 (4.2) | 1,111 (4.3) | 959 (2.9) | 903 (3.5) |
| Diagnosis Related Group Code | | | | | | | | | | | | |
| 469, Major joint replacement with MCC | 1,410 (5.8) | 1,587 (5.8) | 1,684 (5.6) | 1,708 (5.4) | 1,013 (4.4) | 866 (4.5) | 1,040 (4.6) | 772 (4.2) | 555 (1.8) | 375 (1.5) | 428 (1.3) | 355 (1.4) |
| 470, Major joint replacement without MCC | 22,988 (94.2) | 25,723 (94.2) | 28,294 (94.4) | 30,196 (94.6) | 21,756 (95.6) | 18,308 (95.5) | 21,711 (95.4) | 17,811 (95.9) | 31,230 (98.2) | 24,901 (98.5) | 33,016 (98.7) | 25,187 (98.6) |

MCC: Major complication or comorbidity.

Fig 2 showed the relative quarter-year changes in home discharge rates in treated MSAs compared to control MSAs after policy implementation. All types of hospitalizations –TM, MA and Non-Medicare coverage – showed an increase in home discharge rates after program implementation. The supplemental analysis with linear time trends for MA patients showed the same trend (S3 Fig in S1 File).

Table 2, panel A summarized the DID analyses results based on hospitalizations from 2014 to 2017. For the log adjusted length of stay, TM and non-Medicare hospitalizations showed a statistically significant relative decrease after policy implementation. TM patients hospitalized in treated MSAs had 4.0% lower adjusted length of stay compared to those in control MSAs (p < 0.001), and non-Medicare patients in the treated MSAs had 1.0% lower average adjusted length of stay compared to control MSAs after the policy implementation (p < 0.05). In contrast, MA patients in treated MSAs had 2.0% higher average adjusted length of stay compared to control MSAs after the policy implementation (p < 0.05, Table 2, panel A); however, in the supplemental analysis with pre and post-policy time trends, the MA-covered hospitalizations in treated MSAs had 3.0% lower average adjusted length of stay compared to those in control MSAs (p < 0.05, S2 Table in S1 File).

DID results in Table 2, panel B were limited to hospitalizations from 2015 to 2017 as several event study analyses suggested a potential violation of the parallel trend assumption in the

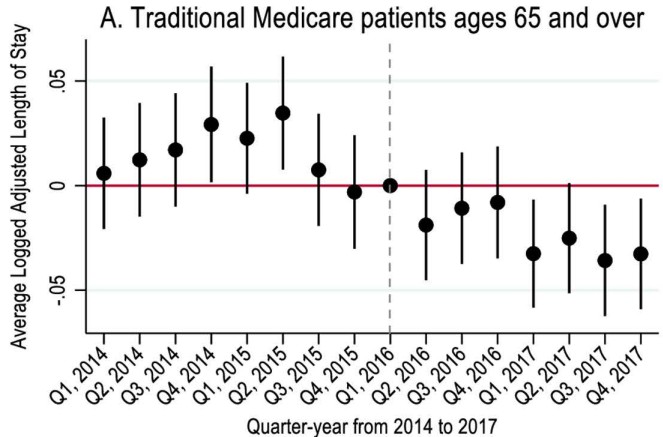

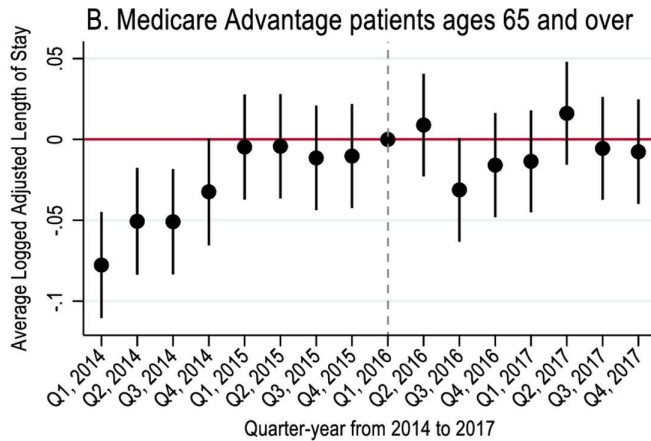

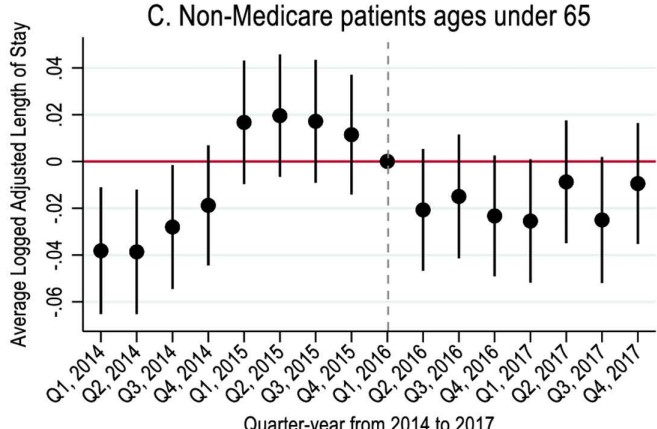

**Fig 1. Change in logged adjusted length of stay of hospitalizations in treatment relative to control MSAs in California from 2014 to 2017 (N = 312,914).**

year 2014. Panel B also showed a relative decrease in adjusted length of stay among TM and non-Medicare hospitalizations in treated MSAs (-4.0% and -3.0%, respectively); however, the relative decrease in MA-covered hospitalizations was no longer significantly different from zero (p > 0.05, Table 2, panel B).

Irrespective of payer, patients in treated MSAs experienced a relative increase in home discharge rates after policy implementation. In Table 2, panel A, TM patients in treated MSAs had 0.02 or 3.4% higher home discharge rates compared to control MSAs after policy implementation (p < 0.001). MA patients in treated MSAs had 0.03 or 4.7% higher home discharge rates, and non-Medicare patients in treated MSAs had 0.02 or 2.3% higher home discharge rates compared to those in control MSAs (p < 0.001). In supplemental analysis, MA patients hospitalized in treated MSAs showed the same trend; however, the coefficient was not significantly different from zero (p > 0.05, S2 Table in S1 File). Table 2, panel B showed similar relative increases in home discharge rates for traditional Medicare, Medicare and non-Medicare patients as in panel A (3.3%, 4.5% and 2.3%, p < 0.05).

In a robustness check of inference for the DID analyses, we employed the wild cluster bootstrap method. The results showed that the relative decrease in adjusted length of stay for TM patients and the relative increase in home discharge rates for MA and non-Medicare patients

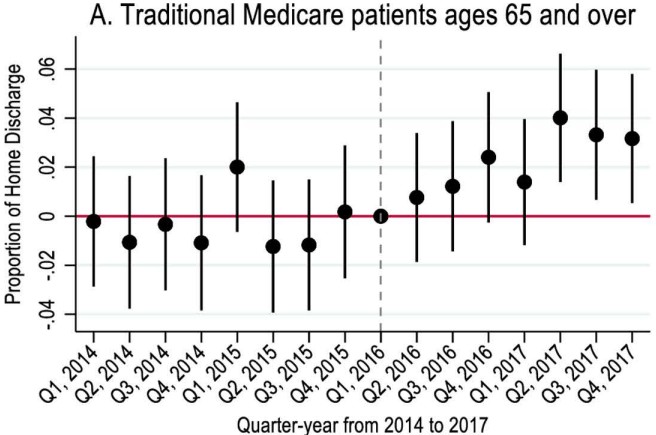

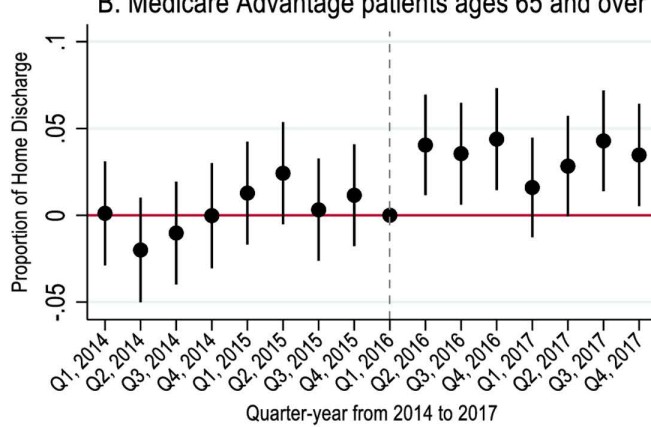

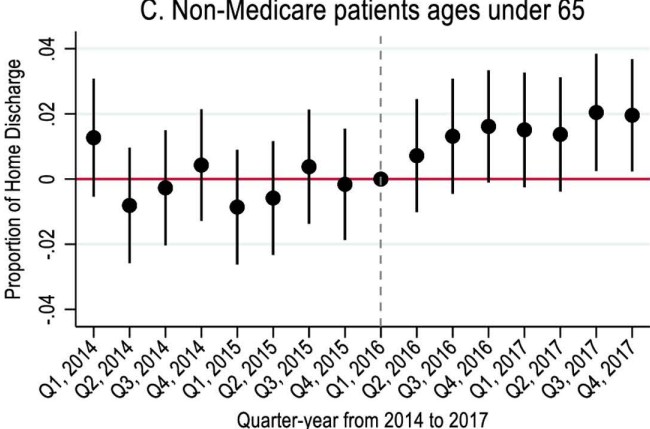

**Fig 2. Change in home discharge rates of hospitalizations in treatment relative to control MSAs in California from 2014 to 2017 (N = 312,914).**

in treated MSAs compared to control MSAs after the policy implementation were still statistically significant after executing 999 replications based on bootstrap samplings with clusters. Changes in adjusted length of stay for MA and non-Medicare patients and home discharge rates for TM became imprecise after clustering (Table 2).

Sensitivity analyses that did not exclude patients based on age yielded similar results as our analytic sample (S3 and S4 Tables in S1 File). In addition, controlling for admission from an ED and Medicaid eligibility did not meaningfully affect the original results (S5 and S6 in S1 File). Employing propensity score weighting to achieve better balance between the treatment and control groups also generated similar results to the original findings (S7 Table in S1 File). The details of the weighting method are provided in the supplemental file.

## Discussion

Medicare's CJR model was designed to reduce costs and improve the quality of care for traditional Medicare patients undergoing total hip and knee replacements. Our study found that the program affected not only traditional Medicare patients but also MA and non-Medicare patients, who were not subject to the payment model. Specifically, the CJR model was associated with a significant decrease in the adjusted length of stay of TM and non-Medicare patients and a significant increase in home discharge rates of TM, MA, and non-Medicare

**Table 2. Comparison of the difference-in-differences analyses results between traditional medicare, medicare advantage and non-medicare patients in treatment and control MSAs in California.**

**Panel A: Study period from 2014 to 2017 (N=312,914)**

| | No. of Hos-pitalizations | Treated MSA | | Control MSA | | Difference in Differences | | | |
|---|---|---|---|---|---|---|---|---|---|
| | | Pre-CJR | Post-CJR | Pre-CJR | Post-CJR | Coefficient (95% CI) | P | Relative Change, % | 95% CI, Wild Cluster Bootstrap |
| Logged Adjusted Length of Stay | | | | | | | | | |
| Traditional Medicare | 113,590 | 1.02 | 0.86 | 0.99 | 0.85 | −0.04 (−0.05, −0.03) | 0.000 | −4.0 | −0.09, −0.01 |
| Medicare Advantage | 83,277 | 0.81 | 0.57 | 0.80 | 0.55 | 0.02 (0.01, 0.03) | 0.002 | 2.0 | −0.16, 0.17 |
| Non-Medicare | 116,047 | 0.78 | 0.56 | 0.72 | 0.54 | −0.01 (−0.02, −0.00) | 0.010 | −1.0 | −0.08, 0.03 |
| Home Discharge Rates | | | | | | | | | |
| Traditional Medicare | 113,590 | 0.58 | 0.67 | 0.61 | 0.68 | 0.02 (0.01,0.03) | 0.000 | 3.4 | −0.00, 0.07 |
| Medicare Advantage | 83,277 | 0.64 | 0.72 | 0.69 | 0.73 | 0.03 (0.02, 0.04) | 0.000 | 4.7 | 0.00, 0.09 |
| Non-Medicare | 116,047 | 0.88 | 0.91 | 0.91 | 0.92 | 0.02 (0.01, 0.02) | 0.000 | 2.3 | 0.00, 0.03 |

**Panel B: Study period from 2015 to 2017 (N=238,326)**

| | No. of Hos-pitalizations | Treated MSA | | Control MSA | | Difference in Differences | | | |
|---|---|---|---|---|---|---|---|---|---|
| | | Pre-CJR | Post-CJR | Pre-CJR | Post-CJR | Coefficient (95% CI) | P | Relative Change, % | 95% CI, Wild Cluster Bootstrap |
| Logged Adjusted Length of Stay | | | | | | | | | |
| Traditional Medicare | 86,812 | 0.99 | 0.86 | 0.96 | 0.85 | -0.04 (-0.05, -0.02) | 0.000 | −4.0 | −0.07, −0.01 |
| Medicare Advantage | 63,830 | 0.75 | 0.57 | 0.73 | 0.55 | -0.00 (-0.02, 0.01) | 0.559 | −0.0 | −0.13, 0.10 |
| Non-Medicare | 87,684 | 0.73 | 0.56 | 0.67 | 0.54 | -0.03 (-0.04, -0.02) | 0.000 | −3.0 | −0.08, 0.00 |
| Home Discharge Rates | | | | | | | | | |
| Traditional Medicare | 86,812 | 0.60 | 0.67 | 0.63 | 0.68 | 0.02 (0.01,0.03) | 0.001 | 3.3 | −0.01, 0.07 |
| Medicare Advantage | 63,830 | 0.66 | 0.72 | 0.70 | 0.73 | 0.03 (0.01, 0.04) | 0.000 | 4.5 | 0.01, 0.05 |
| Non-Medicare | 87,684 | 0.88 | 0.91 | 0.92 | 0.92 | 0.02 (0.01, 0.02) | 0.000 | 2.3 | −0.00, 0.03 |

All the DID analysis was controlled for patient age, age², sex, race/ethnicity, MS-DRG code, and hospital and quarter-year fixed effects. We used default homoskedastic standard errors for our estimation. When logged differences were transformed into percentage changes, the following formula was applied: $\exp(\beta-1) \times 100$.

patients. The spillover effects of the CJR policy on untargeted patient populations were supported by multiple sensitivity checks and robustness analyses. Our most robust finding was that home discharge rates increased for both MA and non-Medicare patients in treated relative to control hospitals after relative to before CJR program implementation.

This study is unique in several ways. First, we used data on all hospitalizations in California and avoided potential bias from datasets that capture only a sample of admissions. Since our dataset included all patients regardless of primary payer, we could examine the spillover effects of the CJR model on all untargeted patients - MA and non-Medicare patients. Second, we estimated both direct and spillover effects of the program in California, which is one of the most diverse states in the United States. Therefore, our study adds to current evidence on the impact of the program, by providing a comprehensive assessment of its effects in a large, varied, and complex healthcare market. Finally, the study contributes to establishing evidence of spillover effects of the program. Our research demonstrates that the program had a positive effect on untargeted patients, particularly on their home discharge rates. The finding of positive spillover effects aligns with and is of similar magnitude to studies that use comprehensive state discharge data [1,11,16] and is inconsistent with studies that rely on only a subset of health care claims data [14,15]. Overall, our study supports the view that value-based traditional Medicare payment models impact health care markets beyond their intended scope.

There are several limitations to our study. First, CJR policy randomization was conducted at the MSA, not at the individual level. This may have introduced bias due to incomplete randomization. Second, as some patient information, such as underlying health conditions, secondary payer sources (e.g., Medicaid for dual eligible beneficiaries), was not available in the dataset, our analyses may over or under-estimate the policy effect due to unobserved differences between patients admitted to treated or control hospitals. To overcome this limitation, we conducted several sensitivity analyses with more saturated models and confirmed that the results were not significantly different from the main analyses. In addition, the study period was limited to 21 months after policy implementation, as CJR changed program participation rules in February 2018. Thus, we could not consider whether spillover effects of the program persisted for many years. Furthermore, because the dataset did not include follow-up information, we could not study outcomes such as readmissions or mortality. Lastly, our study was only based on hospitalizations in California, and our findings may not generalize to other regions.

## Conclusions

In sum, this study demonstrates that the CJR model had significant spillover effects in California. The CJR program affected not only TM patients but was also associated with a substantial decrease in hospital length of stay and an increase in home discharge rates of untargeted Medicare Advantage and non-Medicare patient groups. This finding lends support to studies showing spillover effects of Medicare's value-based payment models.

## Supporting information

**S1 File. S1–7 Tables and S1–3 Figures.**
(PDF)

## Author contributions

**Conceptualization:** Narae Kim.

**Data curation:** Narae Kim.

**Formal analysis:** Narae Kim.

**Investigation:** Narae Kim.

**Methodology:** Narae Kim.

**Supervision:** Mireille Jacobson.

**Validation:** Narae Kim.

**Visualization:** Narae Kim.

**Writing – original draft:** Narae Kim.

**Writing – review & editing:** Mireille Jacobson.

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
