## [Decision Letter · Decision Letter 0]

4 Jan 2024

PONE-D-23-30969The Spillover Effects of Medicare’s Comprehensive Care for Joint Replacement (CJR) Model in CaliforniaPLOS ONE

Dear Dr. Kim,

Thank you for submitting your manuscript to PLOS ONE. After careful consideration, we feel that it has merit but does not fully meet PLOS ONE’s publication criteria as it currently stands. Therefore, we invite you to submit a revised version of the manuscript that addresses the points raised during the review process.

We look forward to receiving your revised manuscript.

Kind regards,

Yu-Chi Tung

Academic Editor

PLOS ONE

Journal Requirements:

"This study was supported by Haynes Lindley doctoral dissertation fellowships from the Haynes Foundation and grant 1R01HS026488-01A1 from the Agency for Healthcare Research and Quality."

3. For studies involving third-party data, we encourage authors to share any data specific to their analyses that they can legally distribute. PLOS recognizes, however, that authors may be using third-party data they do not have the rights to share. When third-party data cannot be publicly shared, authors must provide all information necessary for interested researchers to apply to gain access to the data. (https://journals.plos.org/plosone/s/data-availability#loc-acceptable-data-access-restrictions) 

Reviewers' comments:

Reviewer's Responses to Questions

**Comments to the Author**

1. Is the manuscript technically sound, and do the data support the conclusions?

Reviewer #1: No

Reviewer #2: Yes

2. Has the statistical analysis been performed appropriately and rigorously? 

Reviewer #1: No

Reviewer #2: Yes

3. Have the authors made all data underlying the findings in their manuscript fully available?

Reviewer #1: Yes

Reviewer #2: No

4. Is the manuscript presented in an intelligible fashion and written in standard English?

Reviewer #1: Yes

Reviewer #2: Yes

5. Review Comments to the Author

Reviewer #1: 1 The authors should first define what is the spillover effect.

2 There are many mainstream articles discussing spillover effects, but the authors did not mention any of them. Those mentioned are not CJR related papers.

3 I am wondering since markers of quality of care were not affected, then why do the program still have spillover effects? Of course, I do not have an absolutely right answer. However, the authors should justify this in the introduction section.

4 The authors should apply propensity score matching to ensure that the distribution of every variable in different groups is identical and then present the results in a table.

5 How can the authors make sure there is a 2-year parallel trend before intervention when conducting the DID method? The authors have used some methods to assure parallel trend, but I do not know on which empirical studies the authors base them?

6 It is better to choose readmission, mortality, or infection rates as outcome measures since those three outcomes approximately belong to final outcomes. Moreover, several studies have already addressed them.

I hope these comments are helpful.

Reviewer #2: Dear Authors,

Thank you for the opportunity to review this paper.

This paper analyzed the direct and indirect (spillover) effects of the Comprehensive Care for Joint Replacement (CJR) model on inpatient length of stay and home discharge rates using hospital discharge data from California. Specifically, in order to identify the spillover effects of CJR, authors employed Medicare Advantage (MA) and non-Medicare, who were not affected by CJR. Authors found evidence of spillover effect of CJR.

The major strength of this paper is to contribute to the understanding of the spillover effects of CJR on non-targeted population. Below are some of the specific suggestions (not in order of importance) that can improve the paper.

1. In the study population, authors said that they excluded Medicare beneficiaries who are under age 65 because they might have different medical conditions. Then, I think authors also should exclude beneficiaries eligible for Medicare via End-Stage Renal Disease (ESRD) and Social Security Disability Insurance (SSDI) since they probably have different medical conditions that could bias the estimates.

2. In the outcomes, is there any reason why authors did not include Skilled Nursing Facility (SNF) in the outcome variables? I think it is also important to add SNF because many previous works on CJR showed that CJR is associated with SNF.

3. In the main analysis, did you use clustered standard error? It seems that patients might be subdivided into MSA level.

4. In figure 1 and 2, instead of labelling -9 to 6, it might be better to provide exact year and quarter such as Q1, 2014 and so on.

5. It might be better if authors provide some details (e.g., covariates, standard errors, and so on) in table 2. Readers might be distracted without this information. Also, in table 2, authors should label the second column (e.g., The number of patients or N).

6. PLOS authors have the option to publish the peer review history of their article (what does this mean? ). If published, this will include your full peer review and any attached files.

**Do you want your identity to be public for this peer review?** For information about this choice, including consent withdrawal, please see our Privacy Policy .

Reviewer #1: No

Reviewer #2: No

---

## [Author Response · Author response to Decision Letter 0]

8 Apr 2024

Dear Reviewers,

Thank you for the opportunity to revise our manuscript “The Spillover Effects of Medicare’s Comprehensive Care for Joint Replacement (CJR) Model in California” for re-submission to PLOS ONE.

We have provided a point-by-point response and summary of revisions to each reviewer comment and editorial instruction below and highlighted all changes in the manuscript. We appreciate the reviewers’ feedback and suggestions to strengthen our paper.

Best,

Narae Kim and Mireille Jacobson

Reviewer #1:

1 The authors should first define what is the spillover effect.

Response: Thank you for this feedback. We acknowledge that the audience might not be familiar with thus concept. We added the definition of the spillover effect with an example in the introduction section.

2 There are many mainstream articles discussing spillover effects, but the authors did not mention any of them. Those mentioned are not CJR related papers.

Response: Thank you for this feedback. When we added the definition of spillover effect, we also added some references for readers who would like to learn more about the concept.

3 I am wondering since markers of quality of care were not affected, then why do the program still have spillover effects? Of course, I do not have an absolutely right answer. However, the authors should justify this in the introduction section.

Response: Thank you for your comments. We recognized that the introduction lacked sufficient explanation regarding what constitutes the 'spillover' effect of the program, aside from changes in quality of care. Therefore, we have included the definition of the spillover effects utilized in this study in the introduction: the 'spillover' effects are indirect outcomes of the policy beyond its intended target. In this study, we measured changes in inpatient length of stay and home discharge rates among patients not covered by traditional Medicare to verify the existence of spillover effects from the payment model.

4 The authors should apply propensity score matching to ensure that the distribution of every variable in different groups is identical and then present the results in a table.

Response: Thank you for your suggestions. For our main analysis, we used difference-in-differences methods, which accounts for both observable and unobservable differences that are on similar time paths between treatment and control groups and estimates the relative changes in outcomes of the treated group compared to control group after the policy implementation. Based on your suggestion, however, we acknowledge the concern of composition of treatment and control group systematically changing across time, affecting their trend and biasing the results. Therefore, we conducted additional propensity score weighting as a sensitivity analysis and added the results to our supplemental file. Importantly, the results are very similar when we employ the propensity score method.

5 How can the authors make sure there is a 2-year parallel trend before intervention when conducting the DID method? The authors have used some methods to assure parallel trend, but I do not know on which empirical studies the authors base them?

Response: Thank you for your feedback. We conducted a visual inspection based on unadjusted trend graphs (Supplementary Figure 1 & Supplementary Figure 2) and adjusted pre- and post-policy trend in the event study and DID analyses for MA patients as sensitivity analyses. However, we acknowledge the additional possibility of violation of parallel trend assumption based on several event study analyses results specifically in the year of 2014. Therefore, we limited the study period to 2015 to 2017 and added the results in Table 2 (Table 2 Panel B).

6 It is better to choose readmission, mortality, or infection rates as outcome measures since those three outcomes approximately belong to final outcomes. Moreover, several studies have already addressed them.

Response: Thank you for your suggestion. Unfortunately, our patient discharge dataset, which has detailed information on each hospitalization related to LEJR, has no further follow-up information. Thus, we do not have information regarding readmission, mortality, or infection rates. We discuss this in the limitations section. In addition, we intended to examine how hospitals responded to the policy implementation and how non-traditional Medicare patients were also affected by their behavioral change - patient treatment and care coordination - by measuring current outcomes of interest – length of hospital stay and discharge disposition.

Reviewer #2

1. In the study population, authors said that they excluded Medicare beneficiaries who are under age 65 because they might have different medical conditions. Then, I think authors also should exclude beneficiaries eligible for Medicare via End-Stage Renal Disease (ESRD) and Social Security Disability Insurance (SSDI) since they probably have different medical conditions that could bias the estimates.

Response: Thank you for your feedback. We agree that such conditions could bias our estimation; however, unfortunately, our dataset does not have the information. Instead, we controlled for MSDRG code to adjust for comorbidities and admission from emergency department to control for unobserved health conditions.

2. In the outcomes, is there any reason why authors did not include Skilled Nursing Facility (SNF) in the outcome variables? I think it is also important to add SNF because many previous works on CJR showed that CJR is associated with SNF.

Response: Thank you for your comments. We agree that discharge to SNF is also an important outcome variable dealt in prior studies. However, we decided to examine an increase in home discharge instead of examining a decrease in discharge to SNF as we can compare discharge to home vs. “institutional care,” which includes SNFs.

3. In the main analysis, did you use clustered standard error? It seems that patients might be subdivided into MSA level.

Response: Thank you for your comment. We did not use clustered standard errors at the MSA-level for our main analysis since this method can introduce bias and yield standard errors that are too conservative when the number of treatment clusters is too small. See page 10 for a discussion of this point. Instead, we used wild cluster bootstrap methods to address the concern of observations being clustered at a MSA-level and added the results in the main table (Table 2).

4. In figure 1 and 2, instead of labelling -9 to 6, it might be better to provide exact year and quarter such as Q1, 2014 and so on.

Response: Thank you for your feedback. We changed the labels of figure 1 and figure 2 following your suggestion.

5. It might be better if authors provide some details (e.g., covariates, standard errors, and so on) in table 2. Readers might be distracted without this information. Also, in table 2, authors should label the second column (e.g., The number of patients or N).

Response: Thank you for your feedback. We added the suggested information to Table 2 and label the second column following your suggestion.

---

## [Decision Letter · Decision Letter 1]

6 Sep 2024

PONE-D-23-30969R1The spillover effects of Medicare’s Comprehensive Care for Joint Replacement (CJR) model in CaliforniaPLOS ONE

Dear Dr. Kim,

Thank you for submitting your manuscript to PLOS ONE. After careful consideration, we feel that it has merit but does not fully meet PLOS ONE’s publication criteria as it currently stands. Therefore, we invite you to submit a revised version of the manuscript that addresses the points raised during the review process.

Please address the reviewer's points, including the tables.

We look forward to receiving your revised manuscript.

Kind regards,

Yu-Chi Tung

Academic Editor

PLOS ONE

Journal Requirements:

Reviewers' comments:

Reviewer's Responses to Questions

**Comments to the Author**

1. If the authors have adequately addressed your comments raised in a previous round of review and you feel that this manuscript is now acceptable for publication, you may indicate that here to bypass the “Comments to the Author” section, enter your conflict of interest statement in the “Confidential to Editor” section, and submit your "Accept" recommendation.

Reviewer #1: (No Response)

Reviewer #2: All comments have been addressed

Reviewer #3: (No Response)

2. Is the manuscript technically sound, and do the data support the conclusions?

Reviewer #1: Yes

Reviewer #2: Yes

Reviewer #3: Yes

3. Has the statistical analysis been performed appropriately and rigorously? 

Reviewer #1: No

Reviewer #2: Yes

Reviewer #3: Yes

4. Have the authors made all data underlying the findings in their manuscript fully available?

Reviewer #1: Yes

Reviewer #2: No

Reviewer #3: Yes

5. Is the manuscript presented in an intelligible fashion and written in standard English?

Reviewer #1: Yes

Reviewer #2: Yes

Reviewer #3: Yes

6. Review Comments to the Author

Reviewer #1: 1 It is recommended that relevant tables be included to present the differences in variable distribution before and after matching for patients in the experimental (CJR model) and control groups, in addition to the matching process. For instance, variables such as age could be compared to ensure similarity between the experimental and control groups after matching.

2 Many payment-related literature mentions the spillover effect. It is advised that the authors explore this phenomenon in the introduction. The authors should delineate the academic contributions of their article compared to previous works. For instance:

Glickman, S.W., et al. (2007), in their study titled "Pay for performance, quality of care, and outcomes in acute myocardial infarction," addressed the impact of pay-for-performance mechanisms on healthcare process and outcome, including untargeted measures

Urrusuno, R.F., et al. (2014), in "Compliance with quality prescribing indicators linked to financial incentives: what about not incentivized indicators?: an observational study," focused on the compliance with quality indicators and their association with financial incentives. However, the current study aims to extend this understanding by examining the spillover effects of payment policies on broader healthcare outcomes, thereby contributing to a more comprehensive understanding of payment mechanisms in healthcare.

Reviewer #2: The authors have sufficiently addressed previous comments. As with every study, there are weaknesses, but the authors have acknowledged and addressed those weaknesses. The methodology is sound, and the topic is interesting and valuable.

Reviewer #3: After reviewing the methods and results of this paper, I think the study design is generally appropriate. The use of both DID and event analysis ensures this study presents both the average effect and the micro-, quarterly effect of CJR implementation.

I have some further suggestions for the authors to consider:

1. Results presented should be interpreted carefully.

For example,

“TM patients hospitalized in treated MSAs had 0.04 fewer days or 3.9% lower adjusted length of stay compared to those in control MSAs (p<0.001),..” In this statement, the authors inaccurately interpreted the coefficient on marginal change in logged length of stay as marginal change in days (coefficient is -0.04).

Another example,

“In Table 2A, TM patients in treated MSAs had 0.02 percentage points or 3.4% higher home discharge rates compared to control MSAs after policy implementation (p<0.001).” Is the effect really 0.02 percentage points? If the probability is 0.02, then it should be 2 percentage points, not 0.02 percentage points (note: 0.02 percentage points is really small).

Also, I would recommend the authors to provide more details about how the relative change % was computed and what the meaning of this measure was.

2. To address concerns of clustering, the authors computed Wild Cluster Bootstrap CI. However, some of the results are somehow different from the main analysis. I would recommend authors to speculate reasons for such discrepancy. Did this prove that clustering at MSA level bias the estimates upward?

3. I would recommend the authors check numbers in all the tables (main tables and supplemental) and confirm the accuracy of those estimates. I found some of the DID coefficient does not match with the difference between pre-CJR and post-CJR between groups. In a linear setting (where the marginal effect on length of stay stays in logged form) I would assume the coefficients are differences between “within” and “between” differences.

4. In the Results & Discussion section, the authors call out to Figure S4, but Figure S4 is not available.

5. The add-on propensity score method is interesting. The effect on logged length of stay turns non-significant after weighting. I would suggest the authors to provide a brief description about the PS methods used and how the weighting reduced sample size to 238,326.

7. PLOS authors have the option to publish the peer review history of their article (what does this mean? ). If published, this will include your full peer review and any attached files.

**Do you want your identity to be public for this peer review?** For information about this choice, including consent withdrawal, please see our Privacy Policy .

Reviewer #1: No

Reviewer #2: No

Reviewer #3: No

---

## [Author Response · Author response to Decision Letter 1]

12 Nov 2024

Dear Editor,

We sincerely appreciate your comments. We have revised the manuscript and tried our best to fully address the comments of the reviewers.

Details of the revision are given in the point-by-point response below. In addition, all the changes are tracked in the manuscript.

Warm regards,

Dr. Narae Kim & Dr. Mireille Jacobson

Point-by-point response to reviewers:

Reviewer #1:

1 It is recommended that relevant tables be included to present the differences in variable distribution before and after matching for patients in the experimental (CJR model) and control groups, in addition to the matching process. For instance, variables such as age could be compared to ensure similarity between the experimental and control groups after matching.

Response: Thank you for your comment. Table 1 shows the distribution of relevant variables (e.g., age, racial composition). Please refer to Table 1 for comparisons between groups (treatment vs. control; traditional Medicare vs. Medicare Advantage vs. non-Medicare; before vs. after policy implementation).

It's important to note that even though the distributions of relevant variables differ between treatment and control groups, these differences are not critical to the DID model, which rests on the assumption of similar or “parallel” trends in the pre-treatment period.

2 Many payment-related literature mentions the spillover effect. It is advised that the authors explore this phenomenon in the introduction. The authors should delineate the academic contributions of their article compared to previous works. For instance:

Glickman, S.W., et al. (2007), in their study titled "Pay for performance, quality of care, and outcomes in acute myocardial infarction," addressed the impact of pay-for-performance mechanisms on healthcare process and outcome, including untargeted measures

Urrusuno, R.F., et al. (2014), in "Compliance with quality prescribing indicators linked to financial incentives: what about not incentivized indicators?: an observational study," focused on the compliance with quality indicators and their association with financial incentives. However, the current study aims to extend this understanding by examining the spillover effects of payment policies on broader healthcare outcomes, thereby contributing to a more comprehensive understanding of payment mechanisms in healthcare.

Response: Thank you for your comment. While numerous studies have indeed examined spillover effects in pay-for-performance policies, it's important to note that the focus of Glickman et al. (2007) was primarily on the direct impact of the policy. Their study compared treatment and control hospitals, rather than specifically measuring spillover effects, even though non-targeted outcomes were included in their analysis.

Similarly, Urrusuno et al. (2014) did not concentrate on spillover effects. Instead, their research aimed to demonstrate the impact of financial incentives on prescribing quality indicators. They compared improvements in compliance between incentivized and non-incentivized indicators, concluding that the policy's impact on incentivized indicators was not straightforward.

In contrast, our study aligns more closely with the literature on spillover effects in the Comprehensive Joint Replacement (CJR) program. We've included several CJR studies with conflicting conclusions as our references. Our paper adds value to this ongoing debate by supporting one of these conclusions based on data from California, thus contributing to a more nuanced understanding of spillover effects in healthcare policy, specifically, the CJR program.

Reviewer #2: The authors have sufficiently addressed previous comments. As with every study, there are weaknesses, but the authors have acknowledged and addressed those weaknesses. The methodology is sound, and the topic is interesting and valuable.

Response: We appreciate your comment.

Reviewer #3: After reviewing the methods and results of this paper, I think the study design is generally appropriate. The use of both DID and event analysis ensures this study presents both the average effect and the micro-, quarterly effect of CJR implementation.

I have some further suggestions for the authors to consider:

1. Results presented should be interpreted carefully.

For example,

“TM patients hospitalized in treated MSAs had 0.04 fewer days or 3.9% lower adjusted length of stay compared to those in control MSAs (p<0.001),..” In this statement, the authors inaccurately interpreted the coefficient on marginal change in logged length of stay as marginal change in days (coefficient is -0.04).

Another example,

“In Table 2A, TM patients in treated MSAs had 0.02 percentage points or 3.4% higher home discharge rates compared to control MSAs after policy implementation (p<0.001).” Is the effect really 0.02 percentage points? If the probability is 0.02, then it should be 2 percentage points, not 0.02 percentage points (note: 0.02 percentage points is really small).

Also, I would recommend the authors to provide more details about how the relative change % was computed and what the meaning of this measure was.

Response: Thank you for your comment. We corrected the interpretation of the results – logged length of stay and home discharge rates. Changes are tracked in the manuscript.

2. To address concerns of clustering, the authors computed Wild Cluster Bootstrap CI. However, some of the results are somehow different from the main analysis. I would recommend authors to speculate reasons for such discrepancy. Did this prove that clustering at MSA level bias the estimates upward?

Response: Thank you for your comment. The wild bootstrap does not affect the point estimate but rather the standard errors and thus significance. As we now note in the paper on page 16, this discrepancy suggests that the estimated standard errors from the main analyses were understated, which is a common problem when using clustered standard errors in a setting with few treated clusters.

3. I would recommend the authors check numbers in all the tables (main tables and supplemental) and confirm the accuracy of those estimates. I found some of the DID coefficient does not match with the difference between pre-CJR and post-CJR between groups. In a linear setting (where the marginal effect on length of stay stays in logged form) I would assume the coefficients are differences between “within” and “between” differences.

Response: Thank you for your comment. We checked all the tables and confirmed their accuracy.

4. In the Results & Discussion section, the authors call out to Figure S4, but Figure S4 is not available.

Response: Thank you for your comment. We have correct this and changed it to S3.

5. The add-on propensity score method is interesting. The effect on logged length of stay turns non-significant after weighting. I would suggest the authors to provide a brief description about the PS methods used and how the weighting reduced sample size to 238,326.

Response: Thank you for your comment. The propensity score method was suggested by the other reviewer and added as part of a sensitivity analysis. The weighting did not reduce the sample size; we only applied the method to the 2015-2017 sample (n=238,326). The PS method did not show different results from the original analyses using 2015-2017 data. See Panel B of Table2.

We clarified that the PS method did not change the conclusions from the original analyses in the manuscript (p17) and added the detailed explanation of the PS weighting method we used in the supplemental file (S7 Table).

---

## [Editor Report · Decision Letter 2]

10 Dec 2024

PONE-D-23-30969R2The spillover effects of Medicare’s Comprehensive Care for Joint Replacement (CJR) model in CaliforniaPLOS ONE

Dear Dr. Kim,

Thank you for submitting your manuscript to PLOS ONE. After careful consideration, we feel that it has merit but does not fully meet PLOS ONE’s publication criteria as it currently stands. Therefore, we invite you to submit a revised version of the manuscript that addresses the points raised during the review process.

**Please change the subject “Results & Discussion” to “Results” on page 9.****Why was the total number of hospitalizations (411,004) not equal to the sum (410,978) of the number of hospitalizations from Traditional Medicare (146,662), Medicare Advantage (96,411), and Non-Medicare(167,905) (S1 Table)?****Please delete “log 0.04 or”, “log 0.01 or”, “log 0.02 or”, and “log 0.03 fewer days or” on page 13.****Please move the sentence “This discrepancy suggests that the estimated standard errors from the main analyses were understated, which is a common problem when using clustered standard errors in a setting with few treated clusters (24)” on page 16 to the method section on page 9.**

We look forward to receiving your revised manuscript.

Kind regards,

Yu-Chi Tung

Academic Editor

PLOS ONE
---

## [Author Response · Author response to Decision Letter 2]

26 Jan 2025

Dear Editor,

We sincerely appreciate your comments. We have revised the manuscript and tried our best to fully address the comments of the reviewers.

Details of the revision are given in the point-by-point response below. In addition, all the changes are tracked in the manuscript.

Warm regards,

Dr. Narae Kim & Dr. Mireille Jacobson

Point-by-point response to reviewers:

1. Please change the subject “Results & Discussion” to “Results” on page 9.

Response: We changed the subject accordingly.

2. Why was the total number of hospitalizations (411,004) not equal to the sum (410,978) of the number of hospitalizations from Traditional Medicare (146,662), Medicare Advantage (96,411), and Non-Medicare(167,905) (S1 Table)?

Response: Thank you for the comments. The total population includes patient records (n=26) that lack primary payer information. To address potential confusion, we added a note to the table explaining this issue.

3. Please delete “log 0.04 or”, “log 0.01 or”, “log 0.02 or”, and “log 0.03 fewer days or” on page 13.

Response: We deleted the words accordingly.

4. Please move the sentence “This discrepancy suggests that the estimated standard errors from the main analyses were understated, which is a common problem when using clustered standard errors in a setting with few treated clusters (24)” on page 16 to the method section on page 9.

Response: We moved the sentence accordingly.

---

## [Editor Report · Decision Letter 3]

5 Feb 2025

The spillover effects of Medicare’s Comprehensive Care for Joint Replacement (CJR) model in California

PONE-D-23-30969R3

Dear Dr. Kim,

We’re pleased to inform you that your manuscript has been judged scientifically suitable for publication and will be formally accepted for publication once it meets all outstanding technical requirements.

Kind regards,

Yu-Chi Tung

Academic Editor

PLOS ONE

---

## [Editor Report · Acceptance letter]

PONE-D-23-30969R3

PLOS ONE

Dear Dr. Kim,

I'm pleased to inform you that your manuscript has been deemed suitable for publication in PLOS ONE. Congratulations! Your manuscript is now being handed over to our production team.

Kind regards,

on behalf of

Dr. Yu-Chi Tung

Academic Editor

PLOS ONE